# New Method for Preparing Small-Caliber Artificial Blood Vessel with Controllable Microstructure on the Inner Wall Based on Additive Material Composite Molding

**DOI:** 10.3390/mi12111312

**Published:** 2021-10-26

**Authors:** Junchao Hu, Zhian Jian, Chunxiang Lu, Na Liu, Tao Yue, Weixia Lan, Yuanyuan Liu

**Affiliations:** School of Mechatronics Engineering and Automation, Shanghai University, Shanghai 200444, China; hujunchao@shu.edu.cn (J.H.); jianzhian@shu.edu.cn (Z.J.); 20721813@shu.edu.cn (C.L.); liuna_sia@shu.edu.cn (N.L.); tao_yue@shu.edu.cn (T.Y.); weixia_lan@shu.edu.cn (W.L.)

**Keywords:** small caliber blood vessel, composite molding, micro-nano structure, tissue repair, 3D printing

## Abstract

The diameter of most blood vessels in cardiovascular and peripheral vascular system is less than 6 mm. Because the inner diameter of such vessels is small, a built-in stent often leads to thrombosis and other problems. It is an important goal to replace it directly with artificial vessels. This paper creatively proposed a preparation method of a small-diameter artificial vascular graft which can form a controllable microstructure on the inner wall and realize a multi-material composite. On the one hand, the inner wall of blood vessels containing direct writing structure is constructed by electrostatic direct writing and micro-imprinting technology to regulate cell behavior and promote endothelialization; on the other hand, the outer wall of blood vessels was prepared by electrospinning PCL to ensure the stability of mechanical properties of composite grafts. By optimizing the key parameters of the graft, a small-diameter artificial blood vessel with controllable microstructure on the inner wall is finally prepared. The corresponding performance characterization experimental results show that it has advantages in structure, mechanical properties, and promoting endothelialization.

## 1. Introduction

The incidence and mortality of cardiovascular diseases have been showing an upward trend year by year, which seriously affects human health [1]. Although a large number of vascular stents have been used clinically, most of them are built-in stent products for large blood vessels, and the treatment of small-diameter vascular diseases and functional defects is still a challenge [2,3]. Most of the blood vessels in the cardiovascular and peripheral vascular system are less than 6 mm in diameter [4]. Due to the small inner diameter of such blood vessels and the slow blood flow rate, built-in stents often lead to problems such as thrombosis. Direct replacement with artificial blood vessels is an important potential method [5,6].

The small-caliber blood vessels in the human body are not only small in diameter and thin in wall, but also have a complex layered structure, including an inner layer that supports cells and induces platelet adhesion and aggregation, as well as a middle layer and an outer layer that provide mechanical support [7,8,9,10]. In order to allow the artificial blood vessel to fuse with the host blood vessel after being transplanted into the body, and to quickly achieve the metabolic function of the natural blood vessel, the construction of a small-caliber artificial blood vessel must not only meet the bionic structure and mechanical properties, but also achieve rapid and effective endothelialization. This poses a challenge to both the material design and preparation process [11,12]. A large number of studies have shown that specific microstructures can guide the behavior and arrangement of cells. However, there are few studies that combine this type of research with the construction of small-caliber blood vessels. At the same time, it must be able to take into account the multiple requirements of bionic small-caliber blood vessels in terms of structure and mechanical properties. In addition, in order to obtain a perfect small-caliber artificial blood vessel, the choice of materials is also very important [13,14,15,16]. Because a single material often cannot effectively take into account both biological functions and mechanical properties, natural biological materials combined with polymer composites have become a hot spot in current research [17,18,19]. However, there is still a lack of systematic composite molding process research, and the research on achieving controllable composites of different materials and structures is still very imperfect.

In this context, this research proposes an additive composite molding method that combines electrospinning [20,21,22] and electrostatic direct writing [23] micro-imprint technology. The specific process is as follows. As shown in Figure 1, the first major link includes the following three steps: First, the film structure is obtained by micro-imprinting the material, and, considering the needs of cell adhesion and growth, the film structure can also be imprinted into a surface with a specific microstructure. Then, electrostatically direct wiring forms an orderly arranged fiber structure and transfers the orderly arranged fiber structure to the film structure prepared in advance by embossing. On this basis, the second major link is the process of dynamically shaping the formed two-dimensional film structure into a tube through a tubular mold, which specifically includes the following two steps: one is based on a pre-designed and prepared tubular mold, using thin long tweezers extended from the tail of the mold, and the head stretches out to clamp the previously prepared base film and drag it into the tube. The film will be passively rolled into a tube due to the friction with the tube wall and the boundary effect. The diameter of the tube formed is the inner diameter of the mold. Second, under the drive of the three-axis motion platform, the nanofiber film is directly wrapped on the surface of the above-mentioned dynamically crimped tube through the electrospinning process, which can realize the shape of the dynamically formed tube, and it can also realize the on-demand optimization of the mechanical properties and biological properties of the overall pipe structure.

It should be pointed out that the effective realization of the process method proposed above is closely related to the materials selected and the structural parameters of the pipe to be prepared. First of all, in order to realize that the imprinted film is dynamically shaped into a tube, the mechanical properties of the material and the imprinted film thickness parameters need to be weighed and optimized. For this reason, this article chose a polyether ether ketone (PPDO) material with good biocompatibility, good melting characteristics, and ductility. It is a biodegradable and biocompatible aliphatic polyether ester. It has been approved by the FDA for use, and its degradation is mainly hydrolytic cleavage, which can form low-molecular-weight substances consistent with human metabolites, which can be metabolized or bioabsorbed by the human body [24,25]. Considering the need for cell adhesion growth in the later stage, the PPDO film prepared by imprinting can be surface functionalized by plasma treatment and dopamine soaking. Secondly, the ordered fibers of electrostatic direct writing can be effectively transferred to the imprinted two-dimensional film structure, and it is necessary to comprehensively consider the interface energy competition between the direct writing material, the receiving interface, and the imprinting film. In order to solve this problem, this article proposes that the material selected for electrostatic direct writing and the embossed film material need to have different thermal melt ductility, so that effective transfer can be achieved by adjusting the temperature around the transfer device during the transfer process. In this paper, based on the temperature characteristics of PPDO, combined with the requirements of the electrostatic direct writing process, polycaprolactone (PCL) is selected for electrostatic direct writing. Taking into account the needs of cell adhesion and growth in the later stage, PCL materials can also be compounded with materials with good cell affinity such as gelatin.

It is not difficult to find that the process method proposed in this paper has good processing flexibility and can be controlled in the form of inner wall microstructure, pipe diameter, and wall thickness. The corresponding material selection and specific ratio can also be controlled, adjusted, and designed as needed.

## 2. Materials and Methods

### 2.1. Solution Preparation

Preparation of pcl solution: 2.25 g pcl (Average Mn 80000, Sigma-Aldrich, Co., Haverhill, UK) particles were dissolved in 10 mL dichloromethane (Molecular weight 84.93, XiYaShiJi, Shanghai, China) and 5 mL dimethylformamide DMF (99.5%, Shanghai Aladdin Biochemical Technology Co., Ltd., Shanghai, China), and stirred under a magnetic stirrer (Shanghai MeiYingPu Instrument Manufacturing Co., Ltd., Shanghai, China) for 2 h at a concentration of 15% (*w*/*v*).

Preparation of pcl-gelatin solution: 0.8 g of pcl particles were dissolved in 10 mL of trifluoroethanol (99.5%, Shanghai Aladdin Biochemical Technology Co., Ltd.), and stirred under a magnetic stirrer for 2 h. Subsequently, 0.2 g of gelatin (VetecTM reagent grade, Type A, Sigma-Aldrich, Co., St. Louis, MO, USA) was added and stirred for two hours at 37 degrees, and finally 38 uL of crosslinking agent was added and stirred for 2 h at a concentration of 10% (*w*/*v*).

Dopamine solution preparation: 0.02 g dopamine (Shanghai KeLaMan Reagent Co., Ltd., Shanghai, China) powder was dissolved in 10 mL Tris buffer to prepare a dopamine solution with a concentration of 2 mg/mL.

PPDO: Poly(p-dioxanone) PPDO (Suzhou JIAYE Biotechnology Co., Ltd., Suzhou, China) material is granular. It is a kind of aliphatic polyester ether with good biodegradability and biocompatibility. Its unique ether ester structure gives the material high strength and good flexibility. The melting point is 109 °C. In this experiment, PPDO film was prepared under a hot machine (Qingdao Jinggang hot stamping equipment Co., Ltd., Qingdao, Shandong, China).

### 2.2. Film Fabrication Approaches

#### 2.2.1. Embossed Film Structure Preparation

A plan view of the groove structure is drawn on the drawing software CAD, and the mask plate is processed and prepared. First, a silicon wafer substrate containing a patterned structure is prepared, and the silicon wafer is cleaned then heated on a heating plate at 200 °C for 5 min to remove surface water molecules, followed by spin-coating su-8 2000 glue on the substrate with thick glue spin glue technology. The specific parameters of the rotation speed are: first accelerate to 500 r at an acceleration of 100 r/s and continue for 5 s, then adjust the speed to 2000 r for 30 s, and place it at 95 Heating on a hot plate at 95 °C for 1 min, the thickness of the su-8 2000 adhesive layer is about 0.6 mm.

The mask plate is attached to the adhesive layer and exposed with a lithography machine for 7 s, then heated on a 95 °C heating plate for 1 min, and finally immersed in a developer for 1 min for development and then dried to obtain a pattern-containing mold. The mold is placed in a petri dish, and 28 gpdms (SYLGARDTM 184) and 3 gpdms curing agent are mixed in a beaker and stirred with a glass rod for 3–5 min until milky white and the bubbles are small and uniform. Then it is poured into the petri dish and placed in a vacuum machine for 15 min, then placed in a fume hood for 8 h, and finally baked in an oven at 60 °C for 2 h to obtain a pdms mold with grooved microstructure. Same as above, the steps are repeated to prepare multiple ordinary pdms molds without patterns as auxiliary devices for the imprinting process.

Then, 1 g of PPDO (BaiMuDa, Nanjing, China) particles are placed evenly on the ordinary pdms mold, which are moved to the heating plate with the temperature kept at 120 degrees Celsius. They are heated for 5 min until the PPDO particles melt into a liquid state, and then the whole is transferred to the bottom of the imprinting machine. The PDMS mold containing the groove structure is put on the top to fit it, the air pump control valve adjusted to 0.3 MPa, the imprinting machine control switch turned on, and the hot plate squeezes the PPDO material downwards to adjust the temperature control and time module. The temperature is 120 °C and the duration is 30 min. After cooling, the PPDO base film containing the groove microstructure can be obtained.

#### 2.2.2. Ordered Fiber Structure Preparation

In order to be able to effectively analyze the influence of different material components on subsequent cell behavior, this paper designed two sets of samples, namely the pcl group and the pcl-gelatin group.

The prepared PCL solution is loaded into the syringe piston barrel and connected with the syringe on the micro pump actuator through a catheter. The spinning collector is fixed on the *XY*-axis platform of the three-axis motion platform and the syringe needle and the panel of the spinning collector are made to perpendicularly intersect, adjusting the *Z*-axis slider so that the distance between the end of the syringe needle and the collector is 5 mm. The positive pole of the high-voltage power supply is connected to the metal part of the syringe needle, the negative pole is connected to the metal part of the spinning collector, and the voltage between the two poles is set to 3200 V. The feed flow rate of the micro pump controller is set to 1 mL/h, the reciprocating speed of the needle with the *X*-axis platform of the three-axis motion platform is 0.25 m/s and the single stroke in the positive direction of the *X*-axis is 80 mm. Then, the positive *Y*-axis moves 50 um in the direction, and then moves in the negative direction of the *X*-axis. The distance is 50 um, and the reciprocating movement is repeated many times. With the deposition of pcl on the tin foil of the collector, the tin foil is finally removed to obtain the PCL direct writing fiber structure.

Loading the prepared PCL-gelatin solution into the syringe piston barrel, the flow rate of the micro pump solution is 1 mL/h, the distance between the end of the spinning syringe needle and the collector is 2 mm, and the needle size used is 23 g. The subsequent steps are similar to the previous PCL direct writing steps, and the pcl-gelatin direct writing fiber structure can be obtained.

#### 2.2.3. Preparation of Composite Film

Similar to the steps for preparing the embossed film, firstly, the PPDO base film is prepared by embossing with pdms chips that do not contain microstructures, and the preparation parameters remain the same as above. The PCL direct writing structure is placed together with the tin foil on the base of the imprinting machine. Covering the PPDO base film on the direct writing structure, the air pump control valve is adjusted to 0.3 MPa and the control switch of the imprinting machine is turned on. At this time, the hot plate will squeeze the PPDO film and the direct writing structure downwards. The temperature control and time module are adjusted to 40 °C and the duration is 30 min, and then it is taken out to obtain a composite film of PPDO and direct writing structure.

#### 2.2.4. Preparation of Artificial Blood Vessel

First, a cylindrical through hole with an inner diameter of 4 mm, an outer diameter of 5 mm, and a length of 20 mm is drawn in the solidworks three-dimensional drawing software, and the FDM software is imported to prepare the corresponding mold. Secondly, slender tweezers are used to insert from one end of the mold and extend the other end to clamp the middle of one end of the prepared film and drag it into the tube. Due to the inner wall of the tube, the film spontaneously curls and eventually rolls into a tube. In order to fully discuss the effects of microstructure and materials on cell behavior, the composite films constructed by the above three types of direct-write fibers were respectively crimped, and the film with only embossed groove structure was also selected for crimping. Finally, the three sets of tubular structures prepared above and the mold are connected with a shaft slightly less than 4 mm in diameter, are assembled on a rotating motor, and placed under a three-axis platform for electrospinning. Loading the prepared PCL solution into the syringe piston barrel, it is connected to the micro pump actuator, and the *Z*-axis slider is adjusted so that the distance between the end of the syringe needle and the collector is 100 mm. The voltage between the two poles is set to 7 kV. The feed flow rate of the micro pump controller is set to 1 mL/h, and the internal tubular structure is drawn out by 5 mm for every 5 min of spinning. When all of them are taken out, the blood vessel stent can be obtained; that is, the pcl, pcl-gelatin and PPDO with a groove structure inside the stent.

### 2.3. Characterization of Vascular Grafts

#### 2.3.1. Morphology Observation

In order to observe the guiding structure of the film sample and the macroscopic layered structure of the composite blood vessel, the blood vessels prepared in the imprint group, the pcl direct writing composite group film, and the pcl direct writing composite group were prepared respectively. The morphology of the film sample was detected by an optical microscope to observe the surface structure and morphology. The observation of blood vessels was mainly to observe the layered structure and overall size.

#### 2.3.2. Mechanical Properties

The prepared imprinted film, the film compounded with pcl direct writing fiber, the artificial blood vessel prepared with the imprinted film, and the artificial blood vessel prepared with the film compounded with pcl direct writing fiber were respectively subjected to an axial pull-up test. The prepared film has a size of 20 × 16 mm and a thickness of 0.3 mm, and the corresponding tube is prepared on the basis of the two films of the above specifications. All samples were covered with a pcl electrospun film after preparation, and the electrospinning parameters were all kept the same as described above.

The film and the tube sample are clamped on the universal testing machine. Taking the distance between the two clamps as the initial length, at room temperature, the test piece is stretched at a crosshead speed of 20 mm/min until it breaks. Assuming the incompressibility of the material, and considering the length and cross-sectional area, the load-displacement curve is calculated to determine the stress-strain relationship, using the formula ε = (LF − L1)/L1 to calculate the strain based on the initial length of the specimen (L1) and the tensile specimen length when the force (F) is applied to the specimen (LF). The tensile stress is calculated using σ = F/S, where S is the cross-sectional area of the sample. In this case, the cross-sectional area is calculated as S = tw, where t is the thickness of the stent and w is the width of the sample. Here, when stretched, it is divided into the film group (including the imprinted film and the pcl direct writing composite film) and the tubular graft group (including the imprinted blood vessel and the pcl direct writing composite blood vessel). At the same time, when the film group is stretched, it is divided into straight groove or direct writing structure stretch, and the corresponding tubular components are axial stretch and radial stretch.

#### 2.3.3. Suture Maintains Strength

Suture retention strength (SRS) is commonly used to measure the ability of a suture to adhere a graft to surrounding tissues. A universal testing machine is used to test the suture retention strength. Each is cut to obtain a film sample (length = 20 mm, width = 16 mm). Each sample is clamped in the test device at the edge of the film sample originally located. Using a 5-0 nylon surgical suture (Yangzhou Yuankang Medical Instruments Co., Ltd., Yangzhou, China), the other end of the sample is sutured to a distance of 2 mm from the end. The distance between the two needles is 2 mm. The suture is fixed on the hole in the self-made orifice plate, which is connected to the fixture of the test equipment. The suture is pulled out at an extension rate of 2 mm/s. SRS is calculated by dividing the maximum force recorded before the suture is pulled out by the number of sutures.

#### 2.3.4. In Vitro Cytocompatibility

The cells were selected from the same batch of endothelial cells (Human Umbilical Vein Endothelial Cells, HUVEC), and resuscitated in four bottles. After resuscitation, they were added to each sample dish for culture. Each culture sample dish was pre-added with 3 mL of culture medium (Lifeline Cell Technology, LLC, Lonza, Walkersvile, MD, USA). Finally, it was placed in a 37 degree incubator the culture medium changed every three days. During cell culture, the culture flask is pretreated first; that is, 50 mL gelatin solution with a concentration of 0.1% *w/v* is prepared, filtered with a syringe filter (0.22 u), and then used. Finally, the culture flask is filled with the filtered gelatin solution (10 cm diameter petri dish requires 2 mL of solution), until the bottom of the bottle is covered and left at room temperature for 5 min, and then suck gelatin solution is sucked from the petri dish, which can be used for endothelial cell resuscitation.

In the endothelial cell inoculation experiment, four groups were prepared: the PPDO film group without microstructure, PPDO imprint film group, pcl direct writing compound group, and pcl gelatin direct writing compound group. The sample preparation method of the composite group is as shown in the previous section, which is a composite of pcl or pcl gelatin direct writing structure and unpatterned PPDO base film. The sample size of each group is four, of which the unstructured PPDO film group is mainly used as a control group to compare and observe the influence of groove structure on cell behavior. The pcl direct writing compound group and the pcl gelatin direct writing compound group can be used for comparison and the influence of the material components forming the microstructure on the cell behavior can be analysed.

Before cell inoculation, the four groups of samples were treated with plasma for 90 s, and then the four groups of samples were immersed in dopamine solution, kept in the dark at room temperature for 24 h, then rinsed with deionized water three times, and finally placed in room temperature to air dry. Before cell seeding, all samples were irradiated with ultraviolet light on both sides for 30 min, and sealed and stored in a refrigerator at −20 °C.

The above-mentioned processed samples were inoculated and cultured with cells according to the groups, and the growth of cells on the samples at 1, 3, 7, and 14 days was recorded for each group of cells. The samples that need to observe the results of cell growth were stained with crystal violet to observe the staining and growth of the cells. The specific staining steps are as follows: first, the waste liquid of the sample is aspirated and DPBS buffer is added for washing; then, the DPBS buffer is aspirated, 1 mL of paraformaldehyde solution is added, and it is left for 10–15 min. Then, the paraformaldehyde solution is aspirated and washed using the DPBS solution. Finally, crystal violet solution was added and the sample was immersed, placed on a shaker for 10 min, and finally the crystal violet solution was sucked out and cleaned, and the processed sample was placed under a microscope to observe the cell growth results.

## 3. Results

### 3.1. Morphology of Vascular Grafts

Figure 2 shows the morphological observation results of the vascular graft. Figure 2A is the imprinted film, Figure 2B,C are the observation results under different magnification microscopes, Figure 2D is the pcl direct writing composite group film, Figure 2E,F are the observation results under different magnification microscopes, Figure 2G is the pcl Gelatin direct writing composite film, and Figure 2H,I are the observation results under different magnification microscopes. The inner diameter of the tubular structure is about 4 mm, as shown in Figure 2J. The layered structure of the stent remains intact, and the inner and outer membranes can be clearly seen, as shown in Figure 2K. Figure 2L is a macroscopic view of the composite tube. It can be seen that there is a composite direct writing structure on the inner wall. The thickness of the whole membrane is about 0.3 mm, which is close to the average thickness of human veins of 346 ± 121 um.

### 3.2. Mechanical Properties

The tensile test was carried out on the tensile machine to obtain the stress-strain curve of tensile strength and elongation at break, as shown in Figure 3. Since the difference between the pcl composite group and the pcl gelatin group is mainly in affecting cell growth, they are regarded as the same group for mechanical performance testing. In this experiment, the pcl composite group was selected for testing; the imprinting group with or without microstructures is similar. The situation is regarded as the same group for mechanical performance testing. In this experiment, a sample group with a micro-groove structure was selected for testing. Figure 3A,B are, respectively, the radial tensile and tubular axial tensile stress-strain curves of the film sample, and Figure 3D,E are the radial tensile and tubular axial tensile stress-strain curves of the tubular sample, respectively.

For the film samples, the results of Figure 3A,B show that the tensile strength of the pcl composite group is worse than that of the imprinting group, and the corresponding elongation is not as good as the imprinting group. In the direction perpendicular to the groove, the tensile strength of the pcl composite stent is 18.154 MPa, and the tensile strength of the imprinted stent is 9.800 Mpa; in the direction of the groove, the tensile strength of the pcl composite stent is 27.784 MPa, and the tensile strength of the imprinted stent is 27.784 MPa. The tensile strength is 20.516 Mpa. Therefore, it can be found that both sets of samples meet the mechanical performance requirements of natural blood vessels, however, whether it is the stretching of the straight writing (imprinting) structure or the stretching of the vertical writing (imprinting) structure, the corresponding elongation of the composite group will be larger.

For the tubular sample, the results of Figure 3C,D show that the pcl composite stent and the PPDO imprinted stent have similar mechanical properties, and the imprinting group also has a larger elongation. In the radial direction, the tensile strength of the pcl composite stent is 3.279 MPa, and the tensile strength of the imprinted stent is 3.189 Mpa; in the axial direction, the tensile strength of the pcl composite stent is 4.476 MPa, and the tensile strength of the imprinted stent. The intensity is 6.026 Mpa. Since the mechanical performance parameters of the ideal blood vessel are 2–3 Mpa in the radial direction and 4–6 Mpa in the axial direction, it can be found that the samples of the composite group and the imprint group basically meet the requirements of the ideal blood vessel. Comparing the mechanical properties of the film stretch, a comprehensive comparison shows that the composite blood vessel has a slight increase in mechanical properties while maintaining the mechanical requirements of the natural blood vessel.

The suture force of the graft is used to evaluate the sutureability of the graft implanted in the body. As can be seen in Figure 4, the stitching performance of the pcl composite stent is better than that of the imprinting group. The results show that: the maximum load of the pcl composite group sample is 23.552 N, and a total of four strands of nylon thread are used for testing during the experiment, so the final suture force is 5.89 N; the maximum load of the imprint group sample is 19.420 N, and the final suture force is 4.855 N.

Comprehensively looking at the stretch and stitching data, the composite group and the imprint group have similar mechanical performance test results. In fact, the basic film PPDO plays a major role. In terms of stitching, film-like stretch, and tubular stretch properties, it can be seen that composite stents have certain advantages.

### 3.3. Hydrophilic Results

The contact angle is defined as the intersection of the material, water, and air along the surface of the material and the surface of the water droplet. The angle formed by the line, if the contact angle is greater than 90°, the material is judged to be hydrophobic, the larger the angle, the higher the hydrophobicity; if the contact angle is less than 90°, the material is judged to be hydrophilic, and the smaller the angle is, the material is judged to be hydrophilic. Before the four groups of samples were inoculated with cells, the PPDO base film was plasma treated. It can be seen from Figure 5 that before plasma treatment, the contact angle of the sample was 83°, but after plasma treatment, the hydrophilicity of the sample changed to 62°. The results show that the plasma treatment experiment can effectively improve the hydrophilicity of the PPDO film. Therefore, for cell seeding, such results are positive.

### 3.4. Cell Viability

In order to study the stratified vascular inner layer membrane designed to be suitable for cell growth and cling, and thus its potential use as a component of vascular grafts, we evaluated the inner membranes of four groups of samples in vitro, and evaluated the cells on their respective substrates for activity, proliferation, and morphology. These four groups of samples were plasma treated and immersed in dopamine solution. In the experimental results on the first day, the orientation of the cells was observed (Figure 6). Secondly, in the 14-day experiment, the metabolic activity and proliferation of endothelial cells was observed (Figure 7). The results in Figure 6 show that there is no obvious regularity in cell growth in the unstructured PPDO film group. From the results of the other three groups of experiments, they have a certain effect on cell growth (Figure 6B). The cells in the pcl direct writing composite group have clinging growth in the direction perpendicular to the direct writing structure. Figure 6C,D has a certain direction in the direction of cell growth, and this kind of orientation is the prerequisite for the regular arrangement of cells in various tissues; it also plays an important role in maintaining specific functions [26].

In addition, a macroscopic view of cell growth is shown in Figure 7. After the first day of inoculation, staining, and observation showed that the cells in the unstructured PPDO film group grew densely, while the other three groups also had cell attachment. On the third day, the number of cells in group A decreased, and the cells in group BCD were in a growing state. The results on the 7th day showed that the number of cells in the AD group continued to decline, while the number of cells in the BC group was larger. This difference may be due to the use of gelatin composite materials in group B, which is conducive to cell growth and clinging [27,28]; while group C has more groove structures than group A, which indicates that the introduced microstructure is conducive to cell growth; comparing group D samples with group B, there is less gelatin material, and the number of cells is significantly reduced, which indicates that the gelatin composite material has better biocompatibility. The results on the 14th day showed that the growth of the three groups of ACD cells was not as ideal as that of group B, which also verified the inferences made above.

## 4. Discussion

The preparation of small-diameter vascular grafts remains a challenge. Simulating natural blood vessels should not only consider the requirements of small-diameter and tubular structures when constructing the microstructure surface, but also the flexibility requirements. On the other hand, the preparation of microstructures needs to take into account the thin-walled, layered, and other structures of natural blood vessels. These are essential for simulating natural blood vessels. To solve this problem, we designed a layered small-diameter vascular graft that mimics the structure of human blood vessels: an inner layer suitable for cell adhesion and an outer layer that provides mechanical properties. Here, the inner layer is made by micro-imprinting and electrostatic direct writing technology. The plasma and dopamine treatment of the PPDO base film and the groove structure of the inner wall provide the necessary growth conditions for cell attachment. The outer layer is obtained by electrospinning, which is a technology that can produce a shape and structure similar to the natural extracellular matrix (ECM), which provides mechanical properties for the overall vascular graft. The choice of pcl, gelatin, etc. as raw materials is based on their bioactive compatibility and electrospinning properties. In addition, these two materials have been tested to have good effects as vascular grafts.

In addition to the morphology of the basic structure, good mechanical properties are also necessary for small-diameter vascular grafts. The mechanical properties of an ideal blood vessel are 2–3 Mpa in the radial direction and 4–6 Mpa in the axial direction, and the tensile strength of the vascular graft prepared by us meets the requirements in this respect. Another important mechanical property to consider is suture retention. The experiments in this article show that the mechanical properties of the composite group and the imprint group are similar to each other. In fact, the basic film PPDO plays a major role. It can be seen that composite stents have certain advantages in terms of stitching and film-like tensile properties. In terms of biocompatibility, in vitro cell culture experiments showed good cell compatibility. By recording the cell morphology under different experimental groups and different growth periods, it is shown that the samples with PPDO-based film combined with pcl-gelatin direct writing structure are more in line with the expected cell growth effect. In short, in terms of mechanical properties, the composite group has advantages in stretchability and suture performance, so a comprehensive comparison of composite blood vessels is a prerequisite for preparation. In the in vitro cell culture experiment, additional pcl gelatin direct writing composite group samples were added for control. The experimental results show that the pcl-gelatin group has better biocompatibility. In summary, the pcl gelatin direct writing composite group samples meet our expectations.

## 5. Conclusion and Future Work

In this paper, a small-diameter graft is prepared by a combination of electrostatic spinning, electrostatic direct writing and micro-imprinting, which is mainly divided into imprinting group and composite group. The macroscopic structure of blood vessels is similar to that of natural blood vessels, and the microscopic structure can also achieve the expected effects of cell experiments. In terms of mechanical properties, for tensile properties (including axial and radial), the samples of the composite group and the imprint group basically meet the requirements of ideal blood vessels, and the tensile performance of the composite group is slightly enhanced. In terms of suturing performance, the composite group also has a slight advantage, so a comprehensive comparison of composite blood vessels is a prerequisite for preparation. In the HUVEC in vitro cell culture experiment, additional pcl gelatin direct writing composite group samples were added for control. HUVEC in vitro cell culture experiments, apoptosis and staining showed the compatibility of the graft, indicating that the pcl-gelatin group has better biocompatibility. To sum up, the pcl gelatin direct writing composite group samples meet our expectations, but there are still content that can be supplemented. For example, a variety of different patterns can be prepared on the inner wall of blood vessels to observe cell growth. Or, possibilities include preparing the outer wall of the blood vessel to control its thickness, observing the optimal mechanical properties, etc., which will become part of the continued research work in the future.

## Figures and Tables

**Figure 1 micromachines-12-01312-f001:**
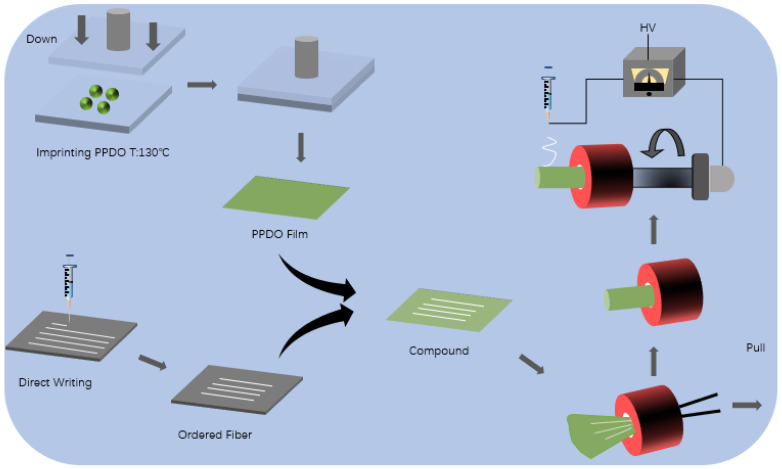
Process flow chart of preparation of small-diameter blood vessels.

**Figure 2 micromachines-12-01312-f002:**
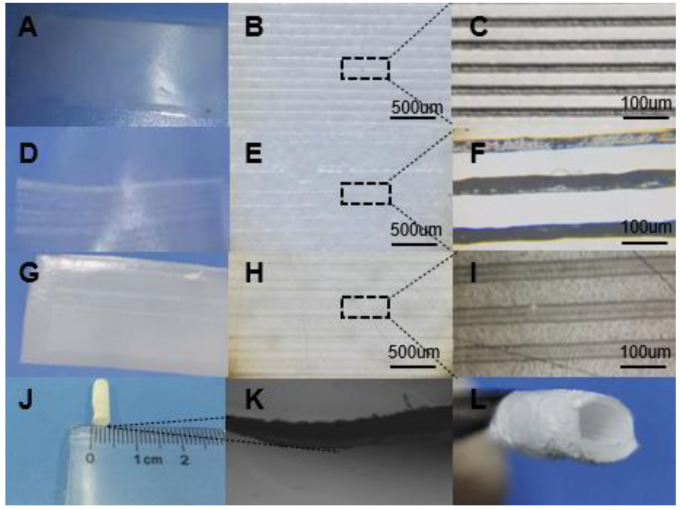
Artificial blood vessel size and film structure: (**A**) is the PPDO imprinting group film structure prepared by imprinting with the pdms chip containing microstructures at 120 °C for 30 min at 120 °C. (**B**,**C**) is the imprinting group film at different magnifications. The microstructure diagram under the microscope (**D**) is a sample film composed of PPDO base film and PCL direct writing structure imprinted with a pdms chip that does not contain microstructures in an imprinting machine at 120 °C for 30 min. The temperature is 40 °C, and the printing time is 30 min. (**E**,**F**) is the microstructure of the PCL direct-write composite film under different magnification microscopes. (**G**) is the imprinting with the pdms chip without microstructure at 120 °C. The sample film composed of the PPDO base film prepared in 30 min and the PCL-gelatin direct writing structure meets the parameters of a temperature of 40 °C and an imprinting time of 30 min (**H**,**I**) for the PCL-gelatin direct writing composite film under different magnification microscopes. The microstructure diagram (**J**) is the overall size of the vascular graft (**K**) is the magnified diagram of the layered structure of the blood vessel (**L**) is the composite diagram of the direct writing structure of the inner wall of the vascular graft.

**Figure 3 micromachines-12-01312-f003:**
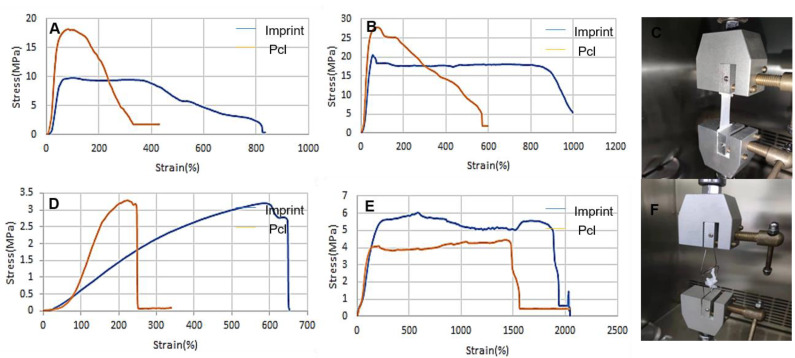
Tensile mechanics test: (**A**) is the drawing of the imprinted film and the pcl direct writing composite film in the vertical groove direction. The tensile strength of the composite stent is 18.154 MPa, and the tensile strength of the imprinted stent is 9.800 Mpa. (**B**) is the drawing of the imprinted film and the pcl direct-writing composite film stretching in the direction of the groove, the tensile strength of the composite stent is 27.784 MPa, and the tensile strength of the imprinted stent is 20.516 Mpa. (**C**) is the film tensile test graph. (**D**) is the drawing of the embossed tubular sample and the pcl direct-write composite tubular sample in the radial direction. The tensile strength of the composite stent is 3.279 MPa, and the tensile strength of the imprinted stent is 3.189 Mpa. (**E**) is the embossed tube. The sample and pcl direct-write composite tubular sample stretched along the axial direction, the tensile strength of the composite stent was 4.476 MPa, and the tensile strength of the imprinted group stent was 6.026 Mpa (**F**) for the tubular tensile experiment.

**Figure 4 micromachines-12-01312-f004:**
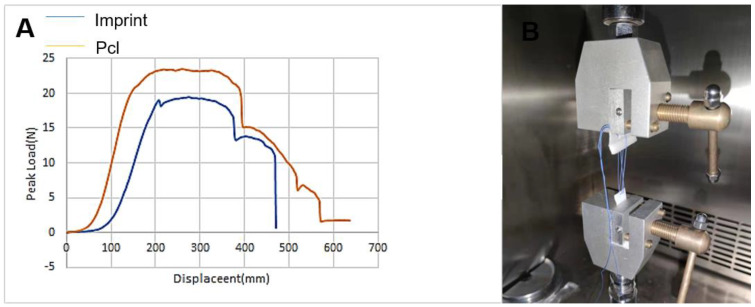
Suture retention experiment: (**A**) is the force_displacement curve of the embossed film and the pcl direct_write composite film suture retention experiment. (**B**) is the schematic diagram of the suture experiment.

**Figure 5 micromachines-12-01312-f005:**
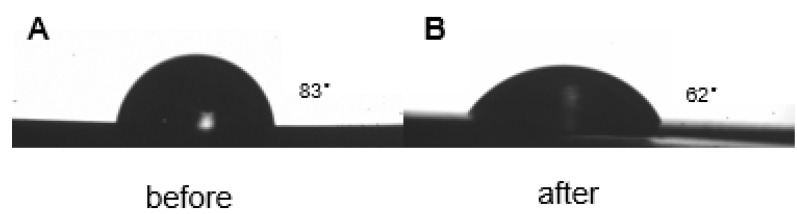
Hydrophilicity of PPDO film before and after plasma treatment: (**A**) before treatment (**B**) after treatment.

**Figure 6 micromachines-12-01312-f006:**
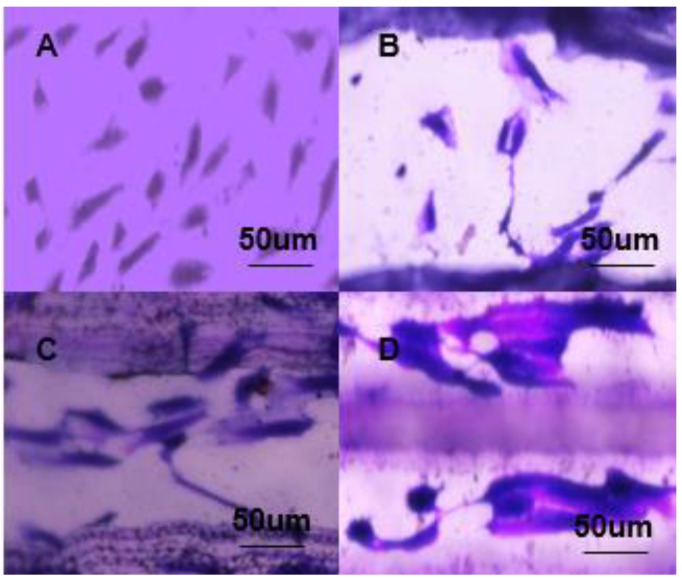
Cell growth on the first day: (**A**) unstructured PPDO film group, (**B**) pcl direct writing compound group, (**C**) pcl gelatin direct writing compound group, and (**D**) PPDO imprinted film group.

**Figure 7 micromachines-12-01312-f007:**
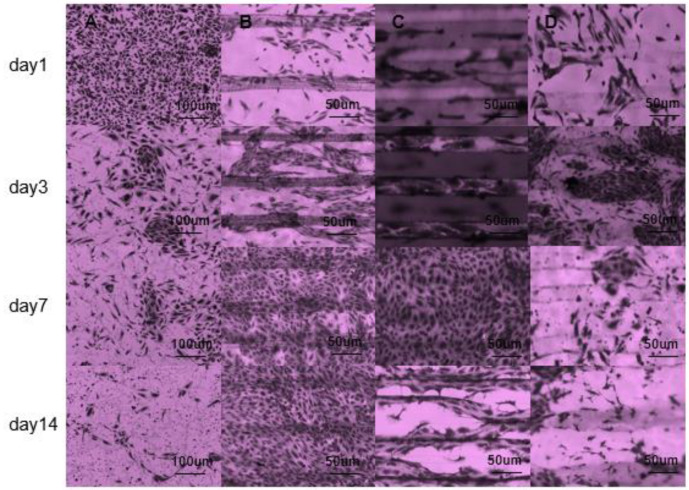
Growth of cells inoculated in two weeks: (**A**) unstructured PPDO film group, (**B**) pcl gelaTable, (**C**) pcl gelatin direct writing compound group, and (**D**) pcl direct writing compound group.

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
