# Peer review of "New Method for Preparing Small-Caliber Artificial Blood Vessel with Controllable Microstructure on the Inner Wall Based on Additive Material Composite Molding"

_micromachines, 2021, doi:10.3390/mi12111312_

Round 1
Reviewer 1 Report
In this work, a layered small-diameter vascular graft that mimics the human vascular structure was developed. This work details the preparation process of the graft and explores the physical properties and biocompatibility of the material. This work has certain practical application value. But there are some major points which need to be addressed in this manuscript to help clarify the data and support the conclusions drawn:
Comment 1: Line 138,139,145 of the manuscript contains obvious errors of expression and handwriting.
Comment 2: In Part 3.2 of this paper, the logic of comparison of tensile strength and corresponding elongation is not clear, please rearrange the data in this part.
Comment 3: The data pictures in Figures 3 and 4 are not clear. Please redraw the data pictures.
Comment 4: In Figure 7, (a) and (b) represent unstructured ppdo film group and pcl gela Table respectively, but (c) and (d) groups do not indicate what they are.
Comment 5: In part 3.4, gelatin can increase cell adhesion of group B. Please cite references to support this conclusion.
Reviewer 2 Report
I propose that:
The aim of the paper is clearly described.
Abstract and conclusion should be rewrite.
Figure 3 and 4 should be clear and scaled on one dimension.
Results is not clearly described.
Controlled technique should be clear defined and must be improved.
This line "The tensile strength and compliance of blood vessels are better than those of natural blood vessels" should be changed. Nothing is better than nature.
